# Crossover between the adiabatic and nonadiabatic electron transfer limits in the Landau-Zener model

Guang Yuan Zhu[1,2], Yi Qin[1,2], Miao Meng [1,2], Suman Mallick[1], Hang Gao[1], Xiaoli Chen[1], Tao Cheng[1], Ying Ning Tan[1], Xuan Xiao[1], Mei Juan Han[1], Mei Fang Sun[1] & Chun Y. Liu [1✉]

The semiclassical models of nonadiabatic transition were proposed first by Landau and Zener in 1932, and have been widely used in the study of electron transfer (ET); however, experimental demonstration of the Landau-Zener formula remains challenging to observe. Herein, employing the Hush-Marcus theory, thermal ET in mixed-valence complexes $\{[Mo_2]-(ph)_n-[Mo_2]\}^+$ ($n = 1$–3) has been investigated, spanning the nonadiabatic throughout the adiabatic limit, by analysis of the intervalence transition absorbances. Evidently, the Landau-Zener formula is valid in the adiabatic regime in a broader range of conditions than the theoretical limitation known as the narrow avoided-crossing. The intermediate system is identified with an overall transition probability ($\kappa_{el}$) of ~0.5, which is contributed by the single and the first multiple passage. This study shows that in the intermediate regime, the ET kinetic results derived from the adiabatic and nonadiabatic formalisms are nearly identical, in accordance with the Landau-Zener model. The obtained insights help to understand and control the ET processes in biological and chemical systems.

[1] Department of Chemistry, Jinan University, 601 Huang–Pu Avenue West, Guangzhou 510632, China. [2]These authors are equally contributed: Guang Yuan Zhu, Yi Qin, Miao Meng. ✉email: tcyliu@jnu.edu.cn

Electron transfer (ET) is a long-standing research subject in chemistry[1-4]. The study leads to better understanding of charge transport in physics and materials science, and the enzymatic redox processes in biology[5,6] and thus, supports the development of modern technologies, including molecular electronics[7] and solar energy conversion[8]. According to the Marcus theory[1,3,5], ET rate is governed by three physical parameters: the Gibbs free energy change ($\Delta G°$), the reorganization energy ($\lambda$) and the electronic coupling (EC) matrix element ($H_{ab}$). The total reorganization energy $\lambda$ is divided into $\lambda_{in}$ and $\lambda_{out}$, corresponding to the intramolecular ($\lambda_{in}$) and solvent ($\lambda_{out}$) nuclear motions. Quantities $\lambda$ and $H_{ab}$ are representative of nuclear and electronic factors, respectively, which affect the ET process through control of the time scales of nuclear motion and electron transition. Both intramolecular and intermolecular ET reactions may occur adiabatically and nonadiabatically, depending on the interplay of the atomic and electronic dynamics of the system and medium. Comparison between the electron hopping frequency ($\nu_{el}$) and nuclear vibrational frequency ($\nu_n$) determines ET in the two regimes, that is[2,9],

$$\text{adiabatic}: \nu_{el} \gg \nu_n; \text{nonadiabatic}: \nu_{el} \ll n_n$$

In the adiabatic limit, ET with concerted nuclear motion proceeds along the ground-electronic-state potential surface constructed based on the Born–Oppenheimer approximation[2,10]. In the nonadiabatic limit, on the other hand, when electron transition takes place, the system turns correspondingly to the final state from the initial state, achieving a "sudden ET"[11].

Nonadiabatic transition of reactions from reactant to product was described first by Landau and Zener in 1930s to describe weakly coupled systems[12,13] in the so-called near-adiabatic regime[14]. By nonadiabatic transition, ET proceeds adiabatically crossing the intersection between the reactant and product potential energy surfaces (PESs), while instantaneous transfer of nuclear amplitude between the two adiabatic states takes place nonadiabatically under the action of nuclear motion. Coupling between the diabatic states of the reactant and the product increases the probability of system traversing the crossing point, eventually leading to ET in the adiabatic limit. The semiclassical Landau–Zener (LZ) model discriminates quantitatively the nonadiabatic and adiabatic limits by three parameters: adiabatic parameter $\gamma$ (Eq. (1)), transition probability $P_0$ (Eq. (2)) with the exponent term being the nonadiabatic transition contribution, and electronic transmission coefficient $\kappa_{el}$ (Eq. (3))[1,2],

$$\gamma = \frac{H_{ab}^2}{2h\nu_n}\sqrt{\frac{\pi}{\lambda k_B T}} \tag{1}$$

$$P_0 = 1 - \exp(-2\pi\gamma) \tag{2}$$

$$\kappa_{el} = 2P_0/(1 + P_0) \tag{3}$$

When $\gamma \gg 1$, the adiabatic limit is realized and for thermal ET $\kappa_{el} \approx 1$, while the nonadiabatic limit prevails with $\gamma \ll 1$ (refs. [1,2,11]). By definition of $\gamma$ (Eq. (1)), it is clear that nonadiabatic transition depends upon the electronic and nuclear factors, represented by $H_{ab}$ and $\nu_n$, respectively. According to the LZ model, thermal ET through nonadiabatic transition takes place in the vicinity of the conical area when the adiabatic avoided crossing is similar to the diabatic crossing. This brings up the general condition that the activation energy ($\Delta G^*$) must be substantially larger than the integral energy ($H_{ab}$), that is, $\Delta G^* \gg H_{ab}$ (refs. [2,14]). With this limit, the LZ formula can be applicable only in a narrow range in terms of EC strength and energy, in contrast to the latter theory, for example, the Zhu–Nakamura theory[15].

The LZ formula has been exploited to predict whether an ET reaction is adiabatic or nonadiabatic; however, experimental manifestation of the theory becomes a challenge. Moreover, identification of the intermediate between the two limits and elucidation of system transformation from one to the other limit are nontrivial, which have been actively explored by theoreticians[2,14,16-18]. Up to now, no experimental study describes the energetic and dynamic features of the intermediate regime. Experimental demonstration and characterizing the intermediate can be possibly accomplished in elemental ET reactions, if an array of systems with the electronic dynamics spanning a broad range of time scales with respect to the nuclear motion is developed. Photoinduced ET is generally in the nonadiabatic regime, while thermal ET occurs usually adiabatically with $\nu_{el} \gg \nu_n$. Testing the LZ model (Eqs. (1–3)) also encounters the technique problems[17]. For example, time-resolved spectroscopy[3] and spectral line-broadening analysis[19] are not appropriate because these methodologies do not provide independent coupling integral ($H_{ab}$) and kinetic parameters as required. Mixed-valence (MV) complexes with two bridged redox sites, generally denoted as a D(donor)–B(bridge)–A(acceptor) assembling, are favorable experimental models due to the properties of the intervalence charge transfer (IVCT) absorption, which measures directly the Franck–Condon barrier of ET ($E_{IT}$) between the electron donor and acceptor[20], which equals to the reorganization energy ($\lambda$) for symmetrical system ($\Delta G° = 0$) based on $E_{IT} = \Delta G° + \lambda$ (refs. [9,11]). Analysis of the IVCT band using the Mulliken–Hush formalism (Eq. (4))[9,21] leads to the coupling energy between the initial and final diabatic states[22].

$$H_{ab} = \frac{2.06 \times 10^{-2}}{r_{ab}} \left( \varepsilon_{IT} \Delta\nu_{1/2} E_{IT} \right)^{1/2} \tag{4}$$

In Eq. (4), $\Delta\nu_{1/2}$ is the IVCT bandwidth at half height and $r_{ab}$ is the effective ET distance. This $H_{ab}$ parameter can be used to calculate the electronic transmission frequency ($\nu_{el}$; Eq. (5)), the adiabatic parameter ($\gamma$; Eq. (1)) and the optical or the thermal ET kinetics based on semiclassical theory at the high temperature limit[5,23].

$$\nu_{el} = \frac{2H_{ab}^2}{h}\sqrt{\frac{\pi^3}{\lambda k_B T}} \tag{5}$$

This approach was first proposed by Taube in 1986 (refs. [24]); unfortunately, it has not succeeded for many decades. The reason for this is that few MV molecular systems exhibit characteristic IVCT bands that allow optical derivations of the ET dynamics and kinetics[9,25,26], although many efforts have been made using the MV systems analogous to the Creutz–Taube ion $[(NH_3)_5Ru(pz)Ru(NH_3)_5]^{5+}$ (refs. [20,24,27]).

Given the characteristic IVCT bands, MV D–B–A molecular systems with a quadruply bonded $Mo_2$ unit[28] as the donor, and a $Mo_2$ unit having a bond order 3.5 as the acceptor are desirable experimental models for study of thermal and optical ET, in which single-electron migration is ensured and the transferring electron is specified to be one of the $\delta$ electrons[29-31]. Here, nine MV complexes of three series with a general formula $[Mo_2(D\text{-}AniF)_3]_2(\mu\text{-}4,4'\text{-}EE'C(C_6H_4)_nCEE')$ (DAniF = $N,N'$-di($p$-anisyl) formamidinate, E, E' = O or S and $n = 1$–3), denoted as $[EE'–(ph)_n–EE']^+$ (Fig. 1), have been synthesized. All these $Mo_2$ dimers exhibit a characteristic IVCT band that varies in transition energy, intensity, and band shape, which provides a desired testbed of the LZ theory through optical analysis by the Mulliken–Hush formalism and ET kinetic study based on the Marcus theory. Complexes $[EE'–(ph)_n–EE']^+$ have small $\lambda_{in}$, as evidenced by the very low IVCT energy for $[SS–ph–SS]^+$ (2650 cm$^{-1}$)[28] in comparison with the Creutz–Taube complex (6369

**Fig. 1 A molecular scaffold for the complexes under investigation.** The three series of [Mo$_2$]–(ph)$_n$–[Mo$_2$] complexes are differentiated by the [Mo$_2$] complex units due to O/S alternation of the chelating atoms (E and E′). Each series consists of three complexes with different (poly)phenylene bridges (ph$_n$, $n$ = 1–3).

cm$^{-1}$)[20]. The adiabaticity of the systems is effectively solvent-controlled because the $\lambda_{in}$ is generally assumed to be independent of the bridge length[32]. This setup of the molecular systems permits to map the parameters $H_{ab}$ and $\lambda$ throughout the adiabatic to the nonadiabatic limits. Incorporating the molecular and electronic dynamics of thermal ET into the LZ model allows the nonadiabatic transition to be examined and the intermediate between the two limits to be characterized. Importantly, our experimental results demonstrate that the LZ formula is practically applicable in a broad range in the adiabatic regime, but not limited by $\Delta G^* \gg H_{ab}$. Two intermediate systems, [OS–(ph)$_3$–OS]$^+$ and [SS–(ph)$_3$–SS]$^+$, are identified with an overall transition probability of ~0.5 that is achieved through operation of the single and the first multiple passage of nuclear motion. Now, we present the experimental demonstration of the LZ model, revealing the energetic and dynamic details of a system crossing over the two limits, which are not well described by this model. With the results from this MV [EE′–(ph)$_n$–EE′]$^+$ system, unification of the contemporary ET theories under the semiclassical framework is visualized[33].

## Results

**Synthesis and characterization of the mixed-valence Mo$_2$ dimers.** Using the published procedure for preparation of the phenylene (ph)- and diphenylene (ph$_2$)-bridged analogs[29,30,34], three terphenylene-bridged Mo$_2$ dimers in [EE′–(ph)$_3$–EE′]$^+$ were synthesized by assembling two dimolybdenum complex units Mo$_2$(DAniF)$_3$(O$_2$CCH$_3$) complexes with a bridging ligand, 4,4′-terphenyldicarboxylate or its thiolate derivatives, 4,4′-(EE′C (C$_6$H$_4$C)$_3$EE′)$^{2-}$ (E, E′ = O or S). The complexes were characterized by $^1$H NMR spectroscopy (Supplementary Figs. 3–5). The solid-state structure of [OS–(ph)$_3$–OS] was determined by X-ray diffraction of a single crystal. The X-ray crystal structure (Supplementary Fig. 7 and Supplementary Table 1) shows that the O and S chelating atoms are arranged in a *trans* manner, as in [OS–ph–OS][34]. The average torsion angle between the neighboring ph groups is ~34°. The centroid distance between the two Mo$_2$ complex units is 20.3 Å, and the edge to edge distance is 14.3 Å, as measured from the C···C distance between the two chelating groups. The Mo$_2$···Mo$_2$ distances for [OO–(ph)$_3$–OO] and [SS–(ph)$_3$–SS] are estimated to be 19.74 and 20.74 Å, respectively, from the crystal structures of the associated complexes in series [EE′–ph–EE′][34].

The MV complexes [EE′–(ph)$_n$–EE′]$^+$ were prepared by one-electron oxidation of the corresponding neutral compounds with one equivalent of ferrocenium hexafluorophosphate[29,30], which were analyzed in situ. These radical cations were characterized by X-band EPR spectra (Supplementary Fig. 8), which exhibit one characteristic signal for $^{96}$Mo ($I = 0$) isotope with some weak hyperfine structures from $^{95}$Mo ($I = 5/2$) and $^{97}$Mo ($I = 5/2$). The EPR peaks center at $g = 1.951$ ([OO–(ph)$_3$–OO]$^+$), 1.953 ([OS–(ph)$_3$–OS]$^+$), and 1.956 ([SS–(ph)$_3$–SS]$^+$), with the $g$ values smaller than that for an organic radical, indicating that the odd electron resides essentially on a $\delta$ orbital[28]. The $g$ values increase as the chelating atoms O are replaced by S atoms, as seen for the ph[29] and ph$_2$ (ref. [30]) series. It is noted that the $g$ values for the ph$_3$ bridged complexes are appreciably large, while smaller $g$ values are obtained for the ph and ph$_2$ series. For these localized MV complexes one would expect smaller $g$ values; increase of the $g$ values implies that the odd electron spends more time on the ph$_3$ bridge.

**Optical behaviors of the mixed-valence complexes.** For the Mo$_2$ dimers, the charge transfer spectra from visible to IR region are pertinent to the $\delta$ electron transition. The MV complexes [EE′–(ph)$_n$–EE′]$^+$ exhibit a metal ($\delta$) to bridging ligand ($\pi^*$) charge transfer (MLCT) absorption in the visible region as the neutral precursors with essentially the same transition energy ($E_{ML}$), but substantially reduced band intensity[29,30,34]. The MLCT band is red shifted with increasing S chelating atoms and blue shifted as the bridge is lengthened (Supplementary Fig. 9 and Table 1)[29,30,34]. For [EE′–(ph)$_n$–EE′]$^+$ with the same ancillary DAniF ligands, the vertical $\delta \rightarrow \delta^*$ transition occurs in a narrow range of wavelengths, ca. 450–500 nm (ref. [28]); however, this band is masked sometimes by the other electronic transitions[29]. For example, careful examination of the spectrum of [OO–(ph)$_3$–OO] found that the absorbance in the 400–600 nm region results from an overlap of the $\delta \rightarrow \delta^*$ transition at 446 nm ($\varepsilon = 8226$ M$^{-1}$ cm$^{-1}$) and the MLCT at 450 nm ($\varepsilon_{ML} = 2853$ M$^{-1}$ cm$^{-1}$; Supplementary Fig. 9A). For [OS–ph–OS]$^+$ and [SS–ph–SS]$^+$, a ligand to metal charge transfer (LMCT) absorption was observed with the transition energy lower than that of the MLCT band[29]. The LMCT band for the MV complexes arises from charge transfer from the $\pi$ orbital of bridging ligand to the $\delta$ orbital of the cationic Mo$_2$ center, thus, corresponding to hole transfer in the opposite direction. Simultaneous presence of the MLCT and LMCT bands facilitates the through-bond superexchange[35], leading to strong EC between the two Mo$_2$ centers[34].

Figure 2 shows the characteristic IVCT bands for serious [EE′–ph–EE′]$^+$ (Fig. 2A) and [SS–(ph)$_n$–SS]$^+$ (Fig. 2B) as the representatives. For [EE′–(ph)$_3$–EE′]$^+$, the IVCT bands in the near-IR region are extremely weak, particularly for [OO–(ph)$_3$–OO]$^+$ (Table 1 and Supplementary Fig. 10). In the Mo$_2$ MV D–B–A systems, upon absorbing low-energy photons ($h\nu = E_{IT}$), vibronic transition (Franck–Condon transition) promotes single ET from the donor (in ground state) to the acceptor (in the excited state). This nonadiabatic ET pathway in [Mo$_2$]–(ph)$_n$–[Mo$_2$], known as optical ET, can be described by Fig. 3.

The ET reaction involves breakage of the $\delta$ bond on the donor and formation of the $\delta$ bond on the acceptor, while the $\sigma$ and $\pi$ bonds remain intact. From the IVCT band, spectral parameters ($E_{IT}$, $\varepsilon_{IT}$, and $\Delta\nu_{1/2}$), are extracted, as listed Table 1, which are used for determination of diabatic coupling energy ($H_{ab}$) using Eq. (2) in the following section. The general variation trends of the IVCT bands for these series are: red shifting of the absorbance with increase of S chelating atoms (Fig. 2A and Table 1) and blue shifting with elongating the bridge (Fig. 2B and Table 1), showing the two factors that affect the EC[29,30]. Thus, the most strongly

**Table 1 Spectroscopic and ET kinetic data[a] and the LZ parameters[b] for mixed-valence complexes [EE′-(ph)$_n$-EE′]$^+$ (E E′ = O or S and n = 1-3).[c]**

|  | [EE′-ph-EE′]$^+$ | | | [EE′-(ph)$_2$-EE′]$^+$ | | | [EE′-(ph)$_3$-EE′]$^+$ | | |
|---|---|---|---|---|---|---|---|---|---|
|  | [OO] | [OS] | [SS] | [OO] | [OS] | [SS] | [OO] | [OS] | [SS] |
| $r_{c\text{-}c}$[d] | 11.24 | 11.7 | 12.24 | 15.44 | 15.9 | 16.44 | 19.74 | 20.2 | 20.74 |
| $E_{IT}$ (cm$^{-1}$) | 4240(12) | 3440(4) | 2650(9) | 8300(8) | 6536(7) | 4830(12) | 12,405(30) | 7406(20) | 6210(15) |
| $\varepsilon_{IT}$ (M$^{-1}$cm$^{-1}$) | 1470(33) | 3690(93) | 12,350(80) | 201(12) | 715(9) | 1614(20) | 52(4) | 224(12) | 315(8) |
| $\Delta\nu_{1/2}$ (cm$^{-1}$) | 4410(63) | 3290(46) | 1766(68) | 5183(60) | 6338(70) | 5231(43) | 3013(46) | 4210(32) | 4426(28) |
| $H_{ab}$ (cm$^{-1}$) | 589(8) | 726(5) | 856(9) | 190(7) | 354(9) | 415(8) | 63(2) | 126(5) | 135(4) |
| $\lambda/4$ (cm$^{-1}$) | 1060(3) | 860(1) | 663(0) | 2075(2) | 1634(2) | 1206(3) | 3101(8) | 1850(5) | 1550(4) |
| $\Delta G^\star$ (cm$^{-1}$) | 581(6) | 287(4) | 83(7) | 1889(4) | 1299(10) | 827(6) | 3038(54) | 1736(12) | 1414(10) |
| ($\lambda/4 - H_{ab}$) | 471(11) | 134(6) | −193(0) | 1885(9) | 1280(11) | 791(11) | 3038(10) | 1724(10) | 1415(8) |
| $k_{et}$(ad) (s$^{-1}$) | 3.0(1) × 10$^{11}$ | 1.4(0) × 10$^{12}$ | 3.4(1) × 10$^{12}$ | 4.1(1) × 10$^{8}$ | 9.3(3) × 10$^{9}$ | 9.2 (3) × 10$^{10}$ | 2.3(3) × 10$^{5}$ | 5.5(1) × 10$^{8}$ | 3.0(2) × 10$^{9}$ |
| $k_{et}$(nonad) (s$^{-1}$) | 7.3(1) × 10$^{11}$ | 3.2(0) × 10$^{13}$ | 1.4(1) × 10$^{14}$ | 4.0(2) × 10$^{8}$ | 1.3(2) × 10$^{10}$ | 1.7(1) × 10$^{11}$ | 2.5(5) × 10$^{5}$ | 5.0(2) × 10$^{8}$ | 2.9(1) × 10$^{9}$ |
| $\nu_{el}$ (s$^{-1}$)[e] | 1.2(1) × 10$^{14}$ | 2.1(1) × 10$^{14}$ | 3.3(1) × 10$^{14}$ | 9.1(4) × 10$^{12}$ | 3.6(2) × 10$^{13}$ | 5.7(2) × 10$^{13}$ | 8.3(5) × 10$^{11}$ | 3.9(4) × 10$^{12}$ | 5.3(4) × 10$^{12}$ |
| $\gamma$ | 1.95 | 3.29 | 5.26 | 0.15 | 0.57 | 0.91 | 0.013 | 0.061 | 0.084 |
| $P_O$ | 1 | 1 | 1 | 0.60 | 0.97 | 1 | 0.076 | 0.32 | 0.41 |
| $\kappa_{el}$ | 1 | 1 | 1 | 0.75 | 0.98 | 1 | 0.14 | 0.48 | 0.58 |

[a]For [EE′-ph-EE′]$^+$ and [EE′-(ph)$_2$-EE′]$^+$, the spectroscopic and ET kinetic Data are cited from refs. [29,30]. Data extraction and analysis are shown in Supplementary Figs. 10–19. In optical analysis, for each complex three independent measurements were taken. Data shown in the parentheses are standard deviations.
[b]The LZ parameters are calculated from Eqs. (1–3).
[c]For all the calculations, an average nuclear vibrational frequency, $\nu_n = 5 \times 10^{12}$ s$^{-1}$, is adopted (ref. [9]).
[d]$r_{c\text{-}c}$ refers to the center to center separation between the two Mo$_2$ centers.
[e]Electronic transition frequencies ($\nu_{el}$) are calculated by Eq. (5).

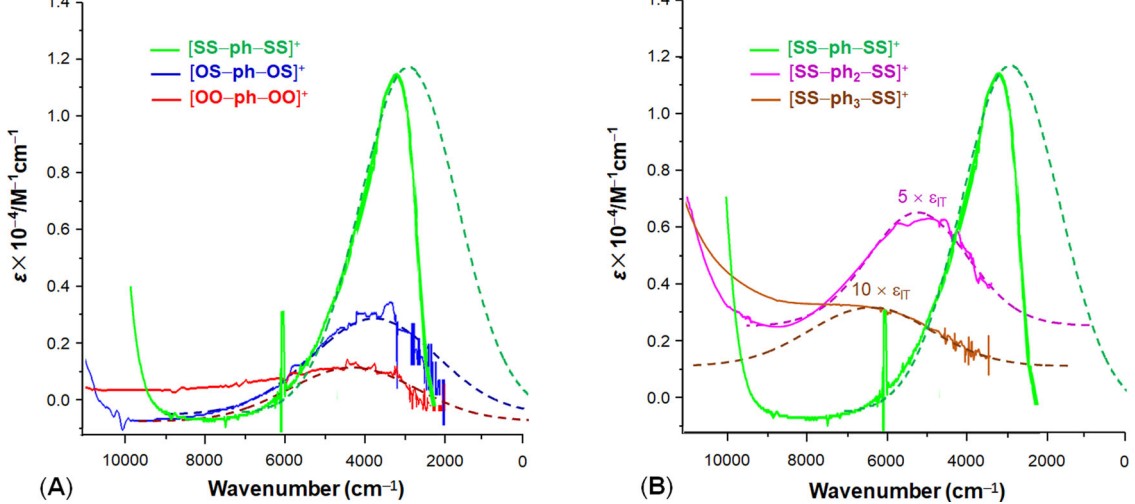

**Fig. 2 Intervalence transition absorptions (IVCT bands) of the MV complexes in the near- to mid-infrared region.** The dashed lines simulate the Gaussian-shaped band profiles to show the spectral asymmetry or spectral "cut-off", which is attributed to strong donor–acceptor electronic coupling. **A** IVCT bands for series [EE′-ph-EE′]$^+$ (EE′ = O or S), showing the spectral characteristics changing with O/S alternation of the chelating atoms on the donor (acceptor). **B** IVCT bands for series [SS-(ph)$_n$-SS]$^+$ (n = 1–3), showing the spectral characteristics changing with variation of the bridge length. For [SS-(ph)$_2$-SS]$^+$ and [SS-(ph)$_3$-SS]$^+$, which are weakly coupled, the IVCT band intensities are magnified by five and ten times for clarity, respectively, because the electronic coupling is very weak. In the spectra, the overtones and vibrational bands in the near-IR and IR regions are trimmed for clarity. The original spectra are presented in the Supplementary Information (Supplementary Figs. 12–21).

coupled [SS-ph-SS]$^+$ exhibits an intense IVCT band in the mid-IR region (Fig. 2), while a high-energy, extremely weak IVCT band is found for [OO-(ph)$_3$-OO]$^+$. It is remarkable that from [SS-ph-SS]$^+$ to [OO-(ph)$_3$-OO]$^+$, the intervalence transition energy increases from 2650 to 12,405 cm$^{-1}$, implicating the great contribution of solvent reorganization energy ($\lambda_{out}$) in controlling the ET dynamics[32]. With the highest reorganization energy (12,405 cm$^{-1}$), [OO-(ph)$_3$-OO]$^+$ is far beyond the solvent-controlled adiabatic regime. In addition, strong coupling endows the MV complex with a asymmetric IVCT band, which is known as the cutting-off phenomenon[36], as shown by [SS-ph-SS]$^+$ and [OS-ph-OS]$^+$ in Fig. 2A. Interestingly, the IVCT bands for [EE′-(ph)$_3$-EE′]$^+$ are narrower than those of the ph$_2$ analogs (Table 1).

This is phenomenal because IVCT band broadening is expected for weaker coupling systems according to $\Delta\nu^0_{1/2} = 2[4\ln(2)\lambda RT]^{1/2}$ (refs. [9,36]). The more delocalized [SS-(ph)$_3$-SS]$^+$ has a $E_{IT}$ comparable to that for the organic MV D–B–A system with a ph$_3$ bridge (6700 cm$^{-1}$)[37].

The EC constants ($H_{ab}$) are calculated from the Mulliken–Hush expression (Eq. (2))[9,21]. In application of Eq. (2) for [EE′-(ph)$_n$-EE′]$^+$, the length of the bridge has been used as the effective ET distance, considering that the δ electrons are fully delocalized over the [Mo$_2$] coordination shell. Therefore, for the ph, ph$_2$, and ph$_3$ series, the geometrical lengths of the bridge "−(C$_6$H$_4$)$_n$−", 5.8, 10.0, and 14.3 Å, respectively, are adopted to be the $r_{ab}$ for the given systems[29,30]. The $H_{ab}$ data are listed in

**Fig. 3 Schematic description of donor–acceptor electron in the phenylene bridged Mo₂ dimer.** In the singly oxidized mixed-valence complex, the quadruply bonded [Mo₂] unit serves as the electron donor and the cationic [Mo₂] unit having a Mo–Mo bond order of 3.5 is the electron acceptor. Electron transfer causes δ bond breakage on the donor and formation on the acceptor, but the other bonds (σ and π) remain intact. While thermal electron self-exchange induced by medium fluctuations occurs between the two dimolybdenum units, nonadiabatic ET undergoes an optical pathway, which proceeds via vibronic transition under the Franck–Condon approximation by absorbing photons (*hv*), exhibiting the IVCT absorption band.

Table 1. Compared to the bridged $d^{5-6}$ metal dimers, the $H_{ab}$ parameters for the Mo₂ MV systems are generally small. The most strongly coupled [**SS–ph–SS**]⁺ has $H_{ab}$ 856 cm⁻¹, smaller than that of the Creutz–Taube ion ($H_{ab} = 1000$ cm⁻¹)[26]. The three series differing in bridge length exhibit a clear variation trend of $H_{ab}$, that is, that $H_{ab}$ decreases with increasing the number of ph group (*n*), as expected from the superexchange pathway[35]. In each series, substitution of S for O on the chelating groups of the bridge increases the $H_{ab}$ value, which can be rationalized by the increased wavefunction amplitude of S atoms that enhances the orbital interaction between the two bridged Mo₂ units. Large decrease of $H_{ab}$, due to the exponential correlation of $H_{ab}$ to $r_{ab}$, is found for the [**EE′–(ph)₃–EE′**]⁺ complexes (Table 1). $H_{ab} = 126$ and 135 cm⁻¹ are determined for [**OS–(ph)₃–OS**]⁺ and [**SS–(ph)₃–SS**]⁺, respectively. For [**OO–(ph)₃–OO**]⁺, the $H_{ab}$ of 63 cm⁻¹ is confirmed by the result ($H_{MM′}$) calculated from the CNS formula (63 cm⁻¹; Supplementary Fig. 17)[25,38], the alternative approach developed by Creutz, Newton, and Sutin. Furthermore, it is worthy of noting that similar $H_{ab}$ values are obtained for [**OS-(ph)₃-OS**]⁺ and [**SS–(ph)₃–SS**]⁺, while the magnitudes of $E_{IT}$ (=λ) are substantially different, implying that the matrix elements are independent of the nuclear geometries for these two systems. It is also interesting to note that among the three series, the variations of $H_{ab}$ resulting from O/S alternation decrease as the bridge length increases. In the ph- and ph₂-bridged series, the differences in $H_{ab}$ between the carboxylate and the fully thiolated analogous are 267 and 225 cm⁻¹, respectively, but in [**EE′–(ph)₃–EE′**], the $H_{ab}$ value increases only 72 cm⁻¹ by the changing atoms E from O to S. These results reflect the diabatic nature of the electronic states in the ph₃ system, in contrast to the adiabatic systems which exhibit the $H_{ab}$ parameters more sensitive to nuclear geometry as shown by the other two series. This phenomenon conforms to the Condon approximation, manifesting a system transition from adiabatic to nonadiabatic with lengthening the bridge, but contradicts the theoretical outcomes with calculated matrix elements[39]. Optical analysis indicates that the two thiolated systems belong to the weak coupling class II, while [**OO–(ph)₃–OO**]⁺ should be assigned to class I, in terms of the Robin–Day's scheme[9,40].

**Electron transfer energetics and dynamics of the mixed-valence systems.** The MV [Mo₂]–bridge–[Mo₂] complex constitutes uniquely an effective "one-particle" donor–acceptor system[11]. In such as a system, adopting a semiclassical two-state LZ model[10,11], the ET initial (φ_I) and final (φ_F) diabatic states can be approximated by the δ orbitals of the donor and acceptor, namely, δ_D and δ_A, respectively. Assuming that the diabatic and

adiabatic states essentially coincide in the vicinity of the electronic equilibrium configurations, linear combinations of δ_D and δ_A generate two first-order or adiabatic states (Eqs. (6) and (7))[31],

$$\Psi_1 = c_a\delta_D + c_b\delta_A = (1/2)^{1/2}(\delta_D + \delta_A) \quad (6)$$

$$\Psi_2 = c_a\delta_D - c_b\delta_A = (1/2)^{1/2}(\delta_D - \delta_A) \quad (7)$$

Then, we have the nonadiabatic mixing matrix element

$$H_{ab} = \langle \delta_D | h | \delta_A \rangle$$

where *h* is an effective one-electron Hamiltonian[11,31]. The energies of the adiabatic states, obtained by solving the two-state secular determinants, are given by Eqs. (8) and (9)[26,36],

$$V_1 = \frac{[\lambda(2X^2 - 2X + 1) + \Delta G^0]}{2} - \frac{\left[(\lambda(2X - 1) - \Delta G^0)^2 + 4H_{DA}^2\right]^{1/2}}{2}$$

$$(8)$$

$$V_2 = \frac{[\lambda(2X^2 - 2X + 1) + \Delta G^o]}{2} + \frac{\left[(\lambda(2X - 1) - \Delta G^o)^2 + 4H_{DA}^2\right]^{1/2}}{2}$$

$$(9)$$

where the reaction coordinate *X* varies from 0 (reactant) to 1 (product) and ΔG° = 0 for the current symmetrical systems. These two adiabatic states are represented by the upper ($V_2$) and lower ($V_1$) PESs, which are separated by $2H_{ab}$ at $X = 0.5$ (refs. [9,11,36]). Study of the strongly coupled systems [**EE′–EE′**]⁺ (*n* = 0) has demonstrated that the upper and lower curves of the adiabatic potential diagram evolve into the electronic energy levels HOMO (δ − δ) and HOMO-1 (δ + δ)[31]. In this strongly coupled limit, [**SS–SS**]⁺, the HOMO–HOMO-1 gap ($\Delta E_{H-H-1}$) equals exactly the measured "IVCT" energy in the spectra and the $2H_{ab}$ calculated from the modified Mulliken−Hush expression for class III system[9,11,36,41], which justify the δ orbitals as the basis of the zero-order wavefunctions of the initial and final diabatic states for the thermal ET for the Mo₂ D–B–A system.

Analysis of the vibronic band gives rise to the λ and $H_{ab}$ (Table 1) for constructions of the adiabatic PESs from Eqs. (8) and (9). Shown in Fig. 4 are the adiabatic PES diagrams for three series, [**OO–(ph)_n–OO**]⁺ (Fig. 4A), [**SS–(ph)_n–SS**]⁺ (Fig. 4B), and [**EE′–(ph)₂–EE′**]⁺ (Fig. 4C). These reaction potential diagrams interpret well the IVCT band characteristics. As shown in Fig. 4A, the three systems in [**OO–(ph)_n–OO**]⁺ present double-well PESs differentiated by the vibronic transition energy ($E_{IT}$) and the adiabatic splitting $2H_{ab}$. [**OO–(ph)₃–OO**]⁺, as the most weakly coupled system, features small curvatures of the diabatic parabolic potential curves. The adiabatic PESs coincide with the diabatic PESs in the conical region with the upper ($V_2$) and lower ($V_1$) surfaces meeting almost at the diabatic crossing point (Fig. 4A). Contrarily, [**OO–ph–OO**]⁺ exhibits a large splitting between the up and low curves at $X = 0.5$. In series [**SS–(ph)_n–SS**]⁺ (Fig. 4B), the PESs for [**SS–ph–SS**]⁺ are dramatically different from those of the systems with longer bridges, although they share a common donor (acceptor). It shows nearly a flat lower $V_1$ surface with two very shallow wells at the reactant and product equilibriums. The separation between $V_1$ and $V_2$ at $X = 0$ corresponds to the low Franck–Condon transition energy ($E_{IT} = \lambda$), close to the adiabatic spacing ($2H_{ab}$) at the avoided crossing (Table 1). The transition state energy ($\Delta G^*$) is only 83 cm⁻¹, much less than the thermal energy level $k_BT$ (207 cm⁻¹ at 298 K). This causes the thermal energy level unevenly populated around the reactant equilibrium; consequently, Franck–Condon transition generates an exact "half cutting-off" IVCT band (Fig. 2)[28,33] typically for class II and III transitional MV systems[26,34,36,42]. For [**SS–(ph)₃–SS**]⁺, on the other hand, the outspreading shift of the

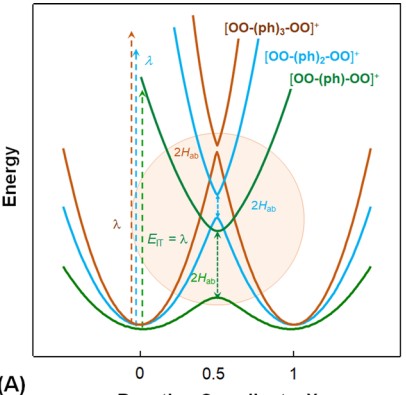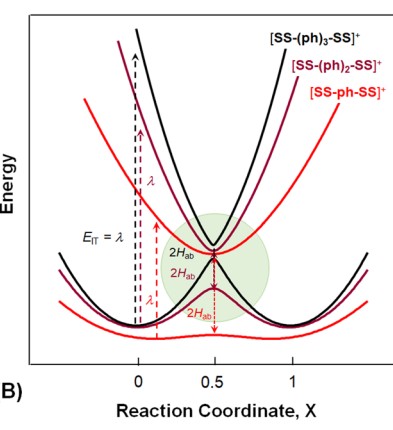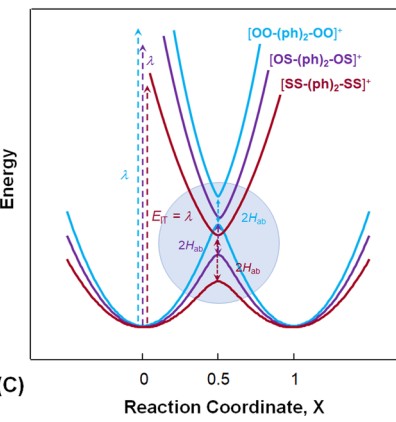

**Fig. 4 Diagrams of the adiabatic potential energy surfaces for the mixed-valence $\{[Mo_2]-(ph)_n-[Mo_2]\}^+$ derived from Eqs. (8) and (9) based on the optically determined $\lambda$ and $H_{ab}$.** For each of the ET system, electronic coupling between the donor and acceptor generates the upper and lower PESs separated by $2H_{ab}$. Vertical transition from the donor equilibrium state to the excited state of the acceptor, represented by a vertical arrow, takes place by absorbing the Frank–Condon energy ($E_{IT}$), which equals numerically the reorganization energy ($\lambda$) in the Marcus theory. **A** Series $[OO-(ph)_n-OO]^+$ ($n=$ 1-3). **B** Series $[SS-(ph)_n-SS]^+$ ($n=$ 1-3). **C** Series $[EE'-(ph)_2-EE']^+$ (E, E' = O or S). For each of the PES diagrams, the avoided crossing area is highlighted.

reactant and product equilibriums and the small curvature of the energy parabola account for the high-energy, narrowed IVCT band, signaling the turning point of system from the solvent-controlled adiabatic to the nonadiabatic regime. Series $[EE'-(ph)_2-EE']^+$ (Fig. 4C), with the same $ph_2$ bridge, shows that the S chelating atoms enhance effectively the EC by lowering $\lambda$ and increasing $2H_{ab}$. In this series, the systems with the same bridge but varied chelating atoms (E) have negligible differences in nuclear organization energy ($\lambda_{in}$); thus, solvent contributions to the ET dynamics and the solvent-controlled adiabaticity are manifested.

According to Marcus[3,5], in the nonadiabatic limit, the thermal activation energy $\Delta G^* = (\lambda + \Delta G^\circ)^2/4\lambda$; in the adiabatic limit, $\Delta G^*$ is reduced by $H_{ab}$ (refs. [9,11]). Since $(\lambda + \Delta G^\circ)^2/4\lambda$ in the nonadiabatic limit is a value of the lowest (i.e., zeroth) order in $H_{ab}$, we can reasonably approximate $\Delta G^*$ by Eq. (10) for the adiabatic–nonadiabatic borderline regime when $H_{ab}$ is sufficiently small[2].

$$\Delta G^* = (\lambda + \Delta G^\circ)^2/4\lambda - H_{ab} \tag{10}$$

For symmetrical system, we have

$$\Delta G^* = \lambda/4 - H_{ab} \tag{11}$$

Then, the difference between $\lambda/4$ and $\Delta G^*$, i.e., $(\lambda/4 - \Delta G^*)$, is expected to equal $H_{ab}$,

$$H_{ab} = \lambda/4 - \Delta G^* \tag{12}$$

These energetic relationships (Eqs. (10)–(12)) show the important correlation between parameters $\Delta G^\circ$, $\Delta G^*$, $\lambda$, and $H_{ab}$ for the transient system, as schematized in Fig. 5A for $[SS-(ph)_3-SS]^+$ and thus, can be used as a quantitative probe of the crossover intermediate. Table 1 lists the values of $(\lambda/4 - H_{ab})$ for each of the systems, in comparison with $\Delta G^*$. Obviously, such a correlation does not exist for strongly coupled systems, for example, $[EE'-ph-EE']^+$ (Table 1). For each series, the deviation between $\Delta G^*$ and $(\lambda/4 - H_{ab})$ decreases as the system nonadiabaticity increases with elongating the bridge. Remarkably, the $(\lambda/4 - H_{ab})$ values, for $[OO-(ph)_2-OO]^+$, $[OS-(ph)_3-OS]^+$, and $[SS-(ph)_3-SS]^+$, are essentially equal to the $\Delta G^*$s. For the most weakly coupled $[OO-(ph)_3-OO]^+$, $\Delta G^*$, and $(\lambda/4 - H_{ab})$ have exactly the same value, 3038 cm$^{-1}$; $(\lambda/4 - \Delta G^*) = 63$ cm$^{-1}$, precisely equaling the $H_{ab}$ (Table 1). These results represent the energetic features of systems in transition from the adiabatic to nonadiabatic limit.

**Electron transfer kinetic study**. The adiabatic ET rate constants, $k_{et}(ad)$, for the MV systems are calculated from the classical transition state formalism[1,5,9] (Eq. 13) with a preexponential factor $\kappa_{el}\nu_n$ and activation energy ($\Delta G^*$) from the Hush–Marcus theory (Eq. (14))[9,41].

$$k_{et} = \kappa_{el}\nu_n \exp\left(-\frac{\Delta G^*}{k_B T}\right) \tag{13}$$

$$\Delta G^* = \frac{(\lambda - 2H_{ab})^2}{4\lambda} \tag{14}$$

The nonadiabatic ET rate constants, $k_{et}(nonad)$, can be determined by the Levich–Marcus expression (Eq. (15))[1,5,23,43]:

$$k_{et} = \frac{2H_{ab}^2}{h}\sqrt{\frac{\pi^3}{\lambda k_B T}}\exp\left(-\frac{\lambda}{4k_B T}\right) \tag{15}$$

For the $[Mo_2]$–bridge–$[Mo_2]$ MV system, the accuracy of the optically determined rate constants is confirmed by IR-band broadening analysis recently[44]. In this work, a transmission coefficient ($\kappa_{el} = 1$–0.14) calculated from the LZ formula (Eqs. (1–3); Table 1) is used to derive $k_{et}(ad)$ from Eq. (13). Given the low-frequency solvent modes $\nu_{out}$ in $10^{12}$–$10^{13}$ s$^{-1}$ in classical theory, an averaged nuclear frequency, $\nu_n = 5 \times 10^{12}$ s$^{-1}$ is generally adopted[9,25]. This is further justified in the present systems in which the nonadiabatic transition is governed by solvent thermal fluctuations. In the nonadiabatic limit, comparison of Eq. (15) to Eq. (13), in conjunction with Eq. (1), gives $\kappa = 2(2\pi\gamma)$ and $\Delta G^* = \lambda/4$, the Marcus activation erengy[5,11]. This indicates implicitly that the adiabatic and nonadiabatic limits are bridged through the intermediate of the LZ model, which can be exploited to test the connection of the existing ET rate expressions in the two limits.

For the ph- and $(ph)_2$-bridged series (Table 1), the electron frequencies ($\nu_{el}$) are in the order of $10^{13}$–$10^{14}$ s$^{-1}$, higher than the nuclear vibrational frequency ($\nu_n$) ($10^{12}$–$10^{13}$ s$^{-1}$) by one order of magnitude. $[SS-ph-SS]^+$ has the highest ET rate with $k_{et}(ad) = 3.4 \times 10^{12}$ s$^{-1}$, close to the adiabatic limit ($5 \times 10^{12}$ s$^{-1}$), in accordance with its optical behavior as a class II and III MV system[26,36,42]. However, the rate constant derived from Eq. (15), $k_{et}(nonad) = 1.4 \times 10^{14}$ s$^{-1}$, is significantly larger than $\nu_n$, indicating the irrationality of the nonadiabatic treatment for this system (Table 1). The deviation of $k_{et}(nonad)$ from $k_{et}(ad)$ decreases with increase of the nonadiabaticity. It is remarkable that for the

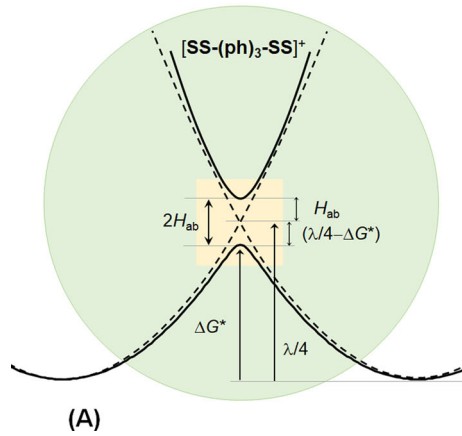
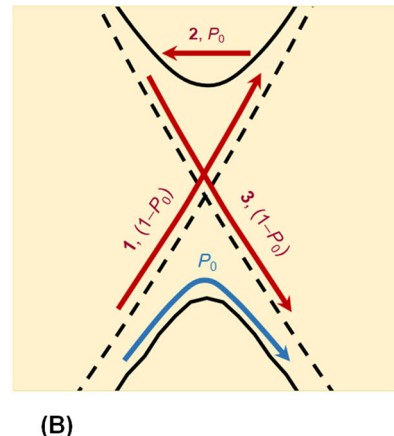

**Fig. 5 Schematic representations of the energetic features of the crossing region and nonadiabatic transition channels for the transient systems. A** The diabatic (dashed line) and adiabatic (solid line) potential energy surfaces in the reactant and product equilibriums and the transition state (shaded area) for [**SS–(ph)$_3$–SS**]$^+$. **B** The first (cyan) and second (red) channels of nonadiabatic transition in intermediate system [**SS–(ph)$_3$–SS**]$^+$. The potential surfaces in **A** and **B** result from zooming in of the avoided crossing area in Fig. 4B. Similar results are expected for [**OS–(ph)$_3$–OS**]$^+$ from the similar $\lambda$ and $H_{ab}$ data.

transient systems, [**OO–(ph)$_2$–OO**]$^+$ and [**EE′–(ph)$_3$–EE′**]$^+$, $k_{et}(ad) = k_{et}(nonad)$ with small analytical errors, and the data fall in the range of $10^8$–$10^9 \, s^{-1}$ (Table 1). Similarly, $k_{et} \sim 10^9 \, s^{-1}$ is reported for the ph$_3$-bridged organic radical system[37]. It is noted that the rate constant for [**SS–(ph)$_3$–SS**]$^+$ is about five times larger than that of [**OS–(ph)$_3$–OS**]$^+$, while their $H_{ab}$ values are similar (Table 1). The high sensitivity of $k_{et}$ on $H_{ab}$ is expected from the increased nonadiabaticity for these systems[1,2,5,17,32]. In contrast, the strongly coupled series [**EE′–ph–EE′**]$^+$ shows $H_{ab}$ independence of the $k_{et}$, (Table 1), in accordance with the theoretical predictions[2,10,11,17,32]. Importantly, the kinetic data demonstrate that the adiabatic and nonadiabatic regimes are smoothly bridged by the crossover regime, which can be well described by the Marcus theory[5,26]. It is surprising that nonadiabatic treatments using solely the average low-frequency nuclear mode ($\nu_n$) on the thermal ET occurring at the intersection of the adiabatic PESs generate precisely consistent outcomes in both the nondiabetic limit and the transient regime. Therefore, this work shows that the adiabatic and nonadiabatic ET rate expressions are applicable in the respective ET dynamic limits, and work equally well with accordant results for the LZ intermediates, although a single theory that rigorously treats the two limits is not available[18,33].

## Discussion

The impacts of $H_{ab}$ and $\lambda$ on $\gamma$ and $\kappa_{el}$ are schematically presented in Fig. 6, which show the smooth systematic transformation from the adiabatic to the nonadiabatic limit. Complexes in [**EE′–ph–EE′**]$^+$ are in the adiabatic limit with $\kappa_{el} = 1$ and $\gamma = 2$–5 ($\gg 1$) due to the short bridge. In [**EE′–(ph)$_2$–EE′**]$^+$, [**SS–(ph)$_2$–SS**]$^+$ has a unity transmission coefficient but the $\gamma$ is lowered to 0.91 (Table 1), while for [**OO–(ph)$_2$–OO**]$^+$, both $\kappa_{el}$ and $\gamma$ are significantly <1. With $\gamma = 0.013$ ($\ll 1$), [**OO–(ph)$_3$–OO**]$^+$ should be placed in the nonadiabatic regime. This is confirmed by the 0.14 $\kappa_{el}$ value, which is close to the nonadiabatic preexponential factor (0.16) calculated from $\kappa = 4\pi\gamma$ (refs. [1,2]). While $\gamma \gg 1$ and $\gamma \ll 1$ characterize the adiabatic and nonadiabatic limits, respectively, the intermediate is not explicitly classified in the LZ model. For [**OS–(ph)$_3$–OS**]$^+$ ($\gamma = 0.061$) and [**SS–(ph)$_3$–SS**]$^+$ ($\gamma = 0.084$), $\gamma$ is much <1, from which the systems might be assigned to the nonadiabatic limit. However, for both, $\kappa_{el} \approx 0.5$, meaning that ~50% of the transition attempts that reach the transition state through thermal fluctuation can successfully complete the ET process. We have seen that these systems present dynamic

and energetic properties in the avoided area that are distinct from those in the adiabatic and nonadiabatic limits. Figure 6 shows clearly the transient status of the systems. Characterization of the intermediate regime is of fundamental importance[15,18]; however, the LZ model[12,13] and other theories[16,17] do not provide such an explicit solution on this issue. According to our results, $\kappa_{el} \approx 0.5$ can be considered to be the practical criterion to probe the transient system. Newton and Sutin pointed out that when $H_{ab} > 200 \, cm^{-1}$, $\kappa_{el} \geq 0.6$ for typical transition metal redox systems[1]. Here, for these two systems with $H_{ab} = 126$ and $135 \, cm^{-1}$, the $\kappa_{el}$ values 0.48 and 0.58 are in excellent agreement with the theoretical predications. For the three [**EE′–(ph)$_3$–EE′**]$^+$ complexes and [**OO–(ph)$_2$–OO**]$^+$, a linear relationship between $\kappa_{el}$ and $\gamma$ is found (Fig. 6C), for which $\gamma < 0.15$, consistent with the theoretic value 0.2 given by Sumi[45]. When $\gamma > 0.5$, $\kappa_{el}$ deviates from the linear dependence on $\gamma$ and approaches unity for $\gamma > 1$, showing the $\gamma$-dependence of $\kappa$ as theoretically predicated[2,45]. Therefore, the experimental results are generally in accordance with theoretic results, but give a narrower and more precise window for $\gamma$ in the correlations between $\kappa_{el}$ and $\gamma$ in the different regimes. For [**OO–(ph)$_3$–OO**]$^+$ in series [**EE′–(ph)$_3$–EE′**]$^+$, the Jortner adiabatic parameter $\kappa_A$ is calculated to be 0.5 (<1), from

$$\kappa_A = \frac{4\pi H_{AB}^2 \langle \tau \rangle}{\hbar \lambda_o} \qquad (16)$$

using $\tau = 1 \, ps$ (ref. [17]), while for the other two, $\kappa_A = 6$–8 (>1), showing the agreement between the two criteria in defining the two ET dynamic limits.

For systems with $P_0 < \kappa_{el}$, involvement of multiple passages in thermal ET reactions is anticipated. For the intermediate systems, it is assumed that two channels, the single passage and the first multiple passage, operate for nonadiabatic transition, as described by Fig. 5B. In the first channel, while the electron makes a transition from the reactant to the product state, the reaction system moves over the crossing point, giving the probability $P_0$. In the second channel, in the course of ET through the avoided area, nuclear motion travers the diabatic crossing point three times to complete the reaction (Fig. 5B)[2]. The first (step 1) and third (step 3) crossing take place on the reactant and product diabatic PESs, respectively, which have the same probability, $(1 - P_0)$. Electron hops from the reactant to the product PES through the second transition (step 2) with the same probability as the first channel ($P_0$). This multiple passage gives the transition

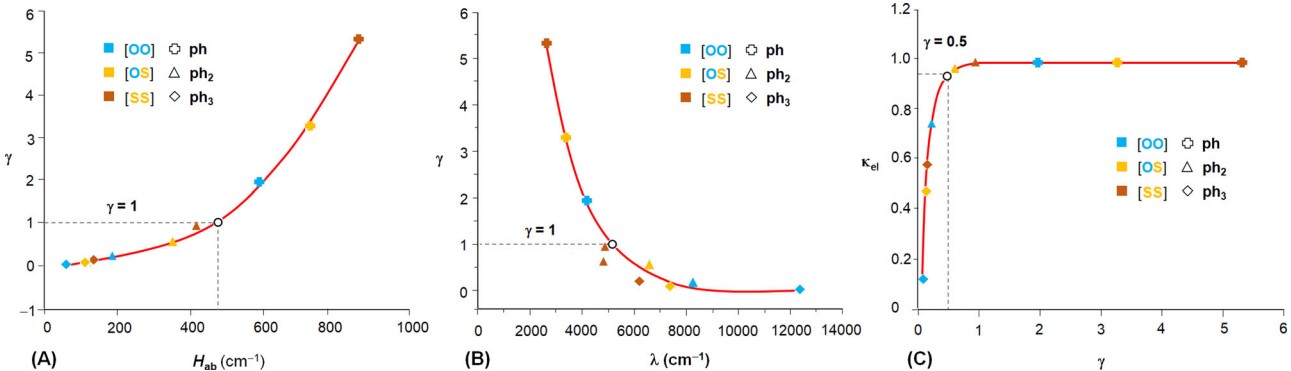

**Fig. 6 Impacts of the electronic and nuclear factors on the Landau–Zener parameters $\gamma$ and $\kappa_{el}$ in systems $[EE'-(ph)_n-EE']^+$ ($E,E' = O$ or $S$, $n = 1-3$). A** Plot of $\gamma$ vs. $H_{ab}$, showing the variation of adiabatic parameter ($\gamma$) as a function of the transfer integral ($H_{ab}$). **B** Plot of $\gamma$ vs. $\lambda$, showing the variation of adiabatic parameter ($\gamma$) as a function of reorganization energy ($\lambda$). **C** Plot of $\kappa_{el}$ vs. $\gamma$, predicting the dependence of transmission coefficient ($\kappa_{el}$) on adiabatic parameter ($\gamma$). Color codes for $[EE'-(ph)_n-EE']^+$: blue for $[OO-(ph)_n-OO]^+$, yellow for $[OS-(ph)_n-OS]^+$, and brown for $[SS-(ph)_n-SS]^+$. The ph, $ph_2$, and $ph_3$ bridges are represented by cross, triangle, and square, respectively.

probability of $(1 - P_0)P_0(1 - P_0)$ ref. [2]. For $[OS-(ph)_3-OS]^+$ and $[SS-(ph)_3-SS]^+$, a transition probability of 0.15 and 0.14 is obtained from the multiple passage, respectively. The total probabilities in this two-channel scheme, ca. 0.48 and 0.55, are close to the overall transmission probabilities ($\kappa_{el}$) 0.48 and 0.58 (Table 1), respectively. This means that for these two systems, 98 and 95% of the successful nonadiabatic hopping events proceed through the first and second channels, with the first channel playing the dominant role. For $[OS-(ph)_3-OS]^+$, the overall transition probability is slightly small because of the relatively weak coupling, in comparison with $[SS-(ph)_3-SS]^+$, but the multiple passage contribution is larger due to the increased nonadiabaticity. It is believed that this two-channel operation can be the typical behavior for thermal ET systems on the adiabatic–nonadiabatic borderline. For $[OO-(ph)_3-OO]^+$, the single and multiple passages are nearly equally important, each of which contributes a transition probability of ~0.07 (Table 1). Evidently, this system has entered the nonadiabatic regime through the intermediate. The small $\kappa_{el}$ (0.14) visulizes the failure of thermal ET through nonadiabatic transition due to the high activation erengy, $\Delta G^* \approx \lambda/4$ (Table 1). However, this does not mean no electron self-exchange occurring between the donor and acceptor. For this long-bridge, weakly coupled MV system, the nonadiabatic ET may occur through optical transition[9], the highly energetic pathway at the same ET rate as for the thermal ET pathway[23]. The multiple trajectory model, developed based on the Fermi Golden rule[33], is the core of the quantum mechanism for nonadiabatic reactions[1,2,45]. Our results show that the single passage is the dominate channel for thermal ET, in accordance with the adiabatic nature of the system.

The LZ model was developed to deal with nonadiabatic coupling in the vicinity of the avoided crossing where $\Delta G \gg 2H_{ab}$. However, the theory does not tell what happens if the kinetic energy is comparable to the interaction energy[2,14]. This is the case represented by $[EE'-ph-EE']^+$, for which $\Delta G^* < 2H_{ab}$. Surprisingly, even for the mostly strongly coupled $[SS-ph-SS]^+$, with the activation energy ($\Delta G^* = 83$ cm$^{-1}$) much smaller than the coupling energy ($H_{ab} = 856$ cm$^{-1}$), the thermal ET can be well described by the LZ parameters, that is, $\gamma = 5.26$ (>1), $P_0 = 1$ and $\kappa_{el} = 1$. Moreover, theoretically, application of the LZ model is limited by the requirement of narrow avoided crossing, that is, that the minimal spacing between the adiabatic PESs at the avoided region should be much smaller than the spacing far from the coupling region[14]. Again, taking $[SS-ph-SS]^+$ as an example, the separations between the surfaces $V_1$ and $V_2$ at the reactant

equilibrium and at the transition state, i.e., 2650 cm$^{-1}$ ($\lambda$) and 1712 cm$^{-1}$ ($2H_{ab}$), respectively, are in the same order of magnitude, which breakdowns the narrow avoided-crossing approximation. Collectively, this study presents a precise scaling picture showing system transition from the adiabatic to nonadiabatic ET limit through the intermediate. The experimental results demonstrate that the LZ formula is practically useful in the ranges of energy and coupling strength that are much broader than the theoretical limits imposed by the nature of the model.

## Methods

**Synthesis**. All manipulations were performed in a nitrogen-filled glove box or by using standard Schlenk-line techniques. All solvents were purified using a vacuum atmosphere solvent purification system or freshly distilled over appropriate drying agents under nitrogen. The phenylene-[29] and biphenylene[30]-bridged Mo$_2$ dimers were synthesized using published procedures. Detailed description of the bridging ligands used in this work is given in the Supplementary Information. The MV complexes used for electron paramagnetic resonant (EPR) and spectroscopic measurements were prepared by one-electron oxidation of the corresponding neutral compounds using 1 equiv. of ferrocenium hexafluorophosphate, of which the spectra were recorded in situ.

**Preparation of $[OO-(ph)_3-OO]$**. A solution of sodium ethoxide (0.014 g, 0.20 mmol) in 10 mL of ethanol was transferred to a solution of Mo$_2$(D-AniF)$_3$(O$_2$CCH$_3$) (0.203 g, 0.20 mmol) in 20 mL of THF. The resultant solution was stirred at room temperature for 0.5 h before the solvents were removed under vacuum. The residue was dissolved in 25 mL of CH$_2$Cl$_2$ and the resultant solution was filtered off through a Celite-packed funnel. The filtrate was mixed with 4,4′-terphenyl dicarboxylic acid (0.0636 g, 0.20 mmol) in 3 mL DMF. The mixture was stirred for 3 h, producing an orange–red solid. The product was collected by filtration and washed with ethanol (3 × 20 mL).
Yield: 0.089 g, 40%.

**General procedure for preparation of $[OS-(ph)_3-OS]$ and $[SS-(ph)_3-SS]$**. A solution of sodium ethoxide (0.033 g, 0.5 mmol) in 5 mL of ethanol was transferred to a solution of Mo$_2$(DAniF)$_3$(O$_2$CCH$_3$) (0.508 g, 0.5 mmol) mixed with either 4,4′-terphenyldithiodicarboxylic acid (0.094 g, 0.27 mmol) for $[OS-(ph)_3-OS]$ or 4,4′-terphenyltetrathiodicarboxylic acid (0.105 g, 0.27 mmol) for $[SS-(ph)_3-SS]$ in THF (30 mL). The respective solutions were stirred at room temperature for 6 h. The solvents were then evaporated under reduced pressure. The residue was dissolved in CH$_2$Cl$_2$ (15 mL) and the solution was filtered through a Celite-packed funnel. The filtrates were concentrated under reduced pressure and the residue was washed with ethanol (3 × 20 mL). The product was collected by filtration and dried under vacuum. Yield of $[OS-(ph)_3-OS]$: 0.305 g, 54%. Yield of $[SS-(ph)_3-SS]$: 0.400 g, 70%.

**Electrochemical characterization**. Electrochemical measurements on the neutral compounds in dichloromethane (DCM) solution were carried out for general evaluation of the EC effect between two Mo$_2$ redox sites. The cyclic voltammograms and differential pulse voltammograms were performed using a CH Instruments model CHI660D electrochemical analyzer in a 0.10 M DCM solution of

$^n$Bu$_4$NPF$_6$ with Pt working and auxiliary electrodes, an Ag/AgCl reference electrode, and a scan rate of 100 mV s$^{-1}$. All potentials are referenced to the Ag/AgCl electrode.

**X-ray structural determination.** Single-crystal data for [**OS–(ph)$_3$–OS**] was collected on a Rigaku XtaLAB Pro diffractometer with Cu-Kα radiation ($\lambda = 1.54178$ Å). Compound [**OS–(ph)$_3$–OS**] crystallized in a monoclinic space group $P2_1/n$ with $Z = 1$. The empirical absorption corrections were applied using spherical harmonics, implemented in the SCALE3 ABSPACK scaling algorithm[46]. The structures were solved using direct methods, which yielded the positions of all non-hydrogen atoms. Hydrogen atoms were placed in calculated positions in the final structure refinement. Structure determination and refinement were carried out using the SHELXS-2014 and SHELXL-2014 programs, respectively[47].

**Electron paramagnetic resonant characterization.** EPR measurements for the MV radicals [**EE′–(ph)$_n$–EE′**]$^+$ were carried out in DCM solution in situ after oxidation at 100 K using a Bruker A300–10–12 EPR spectrometer.

**Spectroscopic measurements.** The electronic (UV–Vis) spectra of the neutral Mo$_2$ dimers [**EE′–(ph)$_n$–EE′**] were recorded on Shimadzu UV-3600 (UV–VIS–NIR) or Cary 600 spectrometer in the range of 300–800 nm. For the MV complexes [**EE′–(ph)$_n$–EE′**]$^+$, to record the low-energy IVCT absorption, a Shimadzu IRAffinity-1s FTIR or Nicolet 6700 FTIR spectrophotometer was used. For those having the main part of the IVCT band extending to the IR region, the spectra were generated by combing the data obtained from the two instruments. All the spectroscopic measurements were conducted in DCM solution ($5 \times 10^{-4}$ mol L$^{-1}$) using quartz cell with light path length of 2 mm.

## Data availability

The X-ray crystallographic data of [**OS–(ph)$_3$–OS**] reported in this study have been deposited at the Cambridge Crystallographic Data Centre (CCDC), under deposition number CCDC 2004426. These data can be obtained free of charge from The Cambridge Crystallographic Data Centre via www.ccdc.cam.ac.uk/data_request/cif. The data that support the findings of this study are available from the corresponding authors upon reasonable request.

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

## Acknowledgements
We are greatly thankful to the National Natural Science Foundation of China (Nos. 21971088 and 21371074), Natural Science Foundation of Guangdong Province (No. 2018A030313894), Jinan University, and Fundamental Research Funds for the Central Universities for the financial support.

## Author contributions
C.Y.L. conceived this project and designed the experiments, and worked on the manuscript. G.Y.Z. and Y.Q. carried out the major experimental work and data analysis. M.M. worked on data collection and analysis of the X-ray crystal structures, and assisted in manuscript work. S.M. participated in data analysis and manuscript preparation, H.G., X. C., X.X., T.C., M.J.H., and M.F.S. were involved in experimental investigation. Y.N.T. performed the DFT calculations on the related complexes.

## Competing interests
The authors declare no competing interests.
