## [Peer Review File · Nature Communications]

REVIEWER COMMENTS

Reviewer #1 (Remarks to the Author):

The manuscript by Liu et al. reports on an experimental validation of the Landau-Zener formula applied to electron transfer processes. The authors use a series of bimetallic mixed-valence complexes to probe the Landau-Zener formula in different regimes of electron transfer. They highlight, in particular, some molecules that belong to an intermediate regime, and show how the Landau-Zener model behaves from the adiabatic to the nonadiabatic limit of electron transfer.

My expertise only allows me to focus here on the more theoretical aspect of this work, and I cannot comment on the experiments conducted and their reliability.

The topic of this article is interesting. Using a series of bimetallic complexes to probe different regimes of electron transfer, compared with results from Landau-Zener theory, sounds like an exciting challenge. However, I have a few general issues with the manuscript in its current form that prevent me from recommending it for publication.

1) The authors state (p.2): "The LZ formula has been exploited to predict whether an ET reaction is adiabatic or nonadiabatic; however, experimental validation of the theory itself has not been achieved". I would somehow disagree with the latter statement, as the Landau-Zener formula is by definition an approximate model for nonadiabatic/adiabatic transitions, with a well-defined derivation (see, for example, 10.1039/C4CP00262H for a recent discussion), and whose limits have been identified in comparison to other strategies for nonadiabatic transitions. More advanced models like the fewest-switches surface hopping by Tully (1990) have been compared to the Landau-Zener model and exact calculations. Hence, I believe that the Landau-Zener model does not need an experimental validation per se due to its inherent approximate nature. My disagreement might be due to a phrasing issue in the paragraph cited above, and I agree that being able to show that the electron-transfer process of complex molecules follows this simple model is in itself very exciting. I found that the discussion related to finding molecules in the intermediate regime, and showing that with the help of the Landau-Zener model, is also appealing. I would advise the authors to modify the text to account for these comments about the nature of the Landau-Zener model. Similarly, the authors also state in p.4 (and likewise in the abstract): "The results and conclusions unify the contemporary ET theories under the semiclassical framework." This statement (and similar ones in the text) should maybe be reformulated considering the body of work by researchers like Joe Subotnik or Noel Hush, who proposed such unifications in different ways and with different methods.

2) I was sometimes left unsatisfied by the discussion of the results presented by the authors. More specifically, I would have appreciated a more detailed analysis of the Hab values trends reported in

Table 1, from a molecular perspective. Such explanations may be trivial for the authors and researchers in the field, but most likely not to the broad readership of Nature Communications.

3) In p.7, the authors discuss H_{ab} and the fact that such matrix elements appear to be independent of the nuclear geometries. They may want to specify that it is a diabatic quantity reflecting the character of electronic states, and therefore by definition less sensitive to nuclear geometries than quantities expressed in the adiabatic representation.

4) I found that the organization and the flow of the text were sometimes not very clear, hampering a proper understanding of the results presented by the authors and how their main findings connect to their conclusions. The last few paragraphs (p.16-17) of the manuscript also suffer from several typographical errors that could have been easily avoided (see list below). Simplifying the structure of the text and maybe leaving some theoretical discussions for the SI might improve the readability of the manuscript if a broad audience is targeted.

Typographical errors:

"Franck-Condon" and "Creutz" are often misspelled in the text.

p.10: "orbital"

p.16: "transition", "avoided", "Surprisingly"

p.17: "requirement", "avoided", "approximation", "nonadiabatic", "demonstrates", "dynamics", "kinetics", "transition".

Caption of Figure 4: a definition of the underlying curve is needed for each plot.

Reviewer #2 (Remarks to the Author):

The paper investigates the electron transfer (ET) process in a systematic series of bridged metal complexes with the goal of validating the Landau-Zener formula for the electron-transfer rate in the non-adiabatic regime.

The paper starts with an account of the basic ingredients in ET theories. Then it proceeds to describe the preparation of the compounds and finally the strategy followed for the extraction of the various parameters.

The main discussion of the paper at the end of page 13, until page 16, compares then the ET rate constants calculated using Marcus theory, which is valid in the adiabatic regime, and LZ rates, and shows how in some cases they provide divergent results, whereas in the intermediate regime they agree quite well.

My impression is that the strength of the paper is the very systematic comparison of many compounds across the adiabatic to non-adiabatic regimes and the use of two different models, Marcus and LZ theories.

I have a more fundamental problem with this work, namely with its premises and how it frames the achieved results:

In line 48 the authors write: "The LZ formula has been exploited to predict whether an ET reaction is adiabatic or nonadiabatic; however, experimental validation of the theory itself has not been achieved."

I have been wondering, what does this statement exactly mean. If the LZ formula had already been used to predict the adiabaticity or lack thereof, it means that the corresponding parameters (for Eqs. 1a,b,c) could already be extracted from experiments before this work, so this part is not really new.

Then I would ask, had the formula made right or wrong predictions?

Specifically, the paper claims that it "validates" the theory. To me, a theory is validated when its prediction matches experiment. Table I lists Marcus and LZ rate constants, but it does not list experimental ET rates measured independently and directly, i.e. not derived from Eqs. 9 and 11.

So, I wonder what has been validated.

Further down in line 301 one reads:

"Therefore, this work shows that the adiabatic and nonadiabatic ET rate expressions are applicable in the respective ET dynamic limits and work equally well with accordant results for the LZ intermediates, although a single theory that rigorously treats the two limits is not available."

Hence, "validation" of the LZ theory means in this work that in the intermediate regime it coincides with another theory that approaches the rate from the other side, and that each one is good in their respective limits. I think this is a very interesting result, but not a validation of the theory per se.

I would suggest that the authors clarify this point and ideally include measured rate constants in Table I to provide an idea about the accuracy of the two theories.

Reviewer #3 (Remarks to the Author):

This is a well-written manuscript that describes the application of the Landau-Zener formula to describe non-adiabatic transitions within the context of thermal electron transfer. The work is based on a series of mixed-valence complexes consisting of pi-bridge coupled Mo₂ dimer units. The experiments are creative and informative, and the topic is highly interesting. The work is likely to be well received and provides a nice explanation of fundamental ET for a wide audience.

The main downside of this paper is the disconnection from the raw spectroscopic data, which would have been better to include at least a couple of representative spectra in the main text, rather than fully relegated to the SI. Perhaps there is a limit on the number of figures available for Nature Communications, but I would strongly encourage the authors to include a figure showing some of the spectra if possible, as it would substantially improve the existing story and also provide a reality check on the level of quantitative detail that may be obtained using this approach.

The other issue I have with the manuscript is one that is actually a strength, overall. The distinction between diabatic and adiabatic modes of ET is somewhat arbitrary. These are really just two different views of the same thing. For example, in eqns 9 and 11, κ is effectively a measure of H_{ab}^2 . In other words, it seems like a circular argument to compare the diabatic and non-adiabatic rate constants that are derived from the same spectroscopic information. Nevertheless, the discussion surrounding diabatic vs adiabatic ET is interesting and forces a closer understanding of the fundamental aspects of ET. The distinction between the two approaches is interesting as it relates to the mechanism for ET, and whether the electron transfer event is sudden and well defined, or occurs gradually (i.e. adiabatically) along the nuclear coordinate. Moreover, the manuscript gives a clear description of the two limiting cases and the data nicely illustrate trends spanning the range between them.

An easy point of confusion is the difference between the terms adiabatic, diabatic, and non-adiabatic. It would be helpful to a broad audience for the authors to explicitly define each of these in the context that they are used here. For example, clarify the statement in line 45 what it means to have “thermal ET though non-adiabatic transition”. At face value, this seems to be ET that occurs from the ground state to an excited state, but here it actually means an adiabatic ET that remains on the lower potential energy curve throughout the reaction, but with a very weakly avoided crossing (i.e. small H_{ab}). (More specifically, it is unclear what to make of the statement: “when the adiabatic aided crossing is similar to the diabatic crossing” on line 46.) While this may be semantics, a clear description will be important to make this interesting work more accessible to a broad audience.

A few minor points:

Line 47 - define ΔG^*

Line 67 (eqn 2) - define $\Delta\nu(1/2)$ and r_{ab}

Line 195 (eqn 5a,b) - define the coordinate, X

Line 200 - HOMO should be the anti-symmetric combination of Mo_2 orbitals (d-d) and HOMO-1 the symmetric combination (d+d)

Line 331 - liner  linear; and delete "in" before "consistent"

Line 339-383 - the final 2 paragraphs have many typos, please edit.

Responses to Referees (NCOMMS-20-29207)

We highly appreciate the reviewers' specialistic, responsible and supportive evaluations on this manuscript. The comments are valuable and helpful to guide us to complete the manuscript revision. In the revised manuscript, all the changes are color highlighted. Please note that a new figure (Fig. 2) is inserted and Fig. 3 is modified. Here we response the reviewer's comments point by point. We are trying to address all the raised issues clearly and in detail. If there still are incorrect statements or further clarification needed, please advise.

Reviewer #1:

The manuscript by Liu et al. reports on an experimental validation of the Landau-Zener formula applied to electron transfer processes. The authors use a series of bimetallic mixed-valence complexes to probe the Landau-Zener formula in different regimes of electron transfer. They highlight, in particular, some molecules that belong to an intermediate regime, and show how the Landau-Zener model behaves from the adiabatic to the nonadiabatic limit of electron transfer. My expertise only allows me to focus here on the more theoretical aspect of this work, and I cannot comment on the experiments conducted and their reliability.

The topic of this article is interesting. Using a series of bimetallic complexes to probe different regimes of electron transfer, compared with results from Landau-Zener theory, sounds like an exciting challenge. However, I have a few general issues with the manuscript in its current form that prevent me from recommending it for publication.

1) The authors state (p.2): "The LZ formula has been exploited to predict whether an ET reaction is adiabatic or nonadiabatic; however, experimental validation of the theory itself has not been achieved". I would somehow disagree with the latter statement, as the Landau-Zener formula is by definition an approximate model for nonadiabatic/adiabatic transitions, with a well-defined derivation (see, for example, 10.1039/C4CP00262H for a recent discussion), and whose limits have been identified in comparison to other strategies for nonadiabatic transitions. More advanced models like the fewest-switches surface hopping by Tully (1990) have been compared to the Landau-Zener model and exact calculations. Hence, I believe that the Landau-Zener model does not need an experimental validation per se due to its inherent approximate nature. My disagreement might be due to a phrasing issue in the paragraph cited above, and I agree that being able to show that the electron-transfer process of complex molecules follows this simple model is in itself very exciting. I found that the discussion related to finding molecules in the intermediate regime, and showing that with the help of the Landau-Zener model, is also appealing. I would advise the authors to modify the text to account for these comments about the nature of the Landau-Zener model. Similarly, the authors also state in p.4 (and likewise in the abstract): "The results and conclusions unify the contemporary ET theories under the

semiclassical framework." This statement (and similar ones in the text) should maybe be reformulated considering the body of work by researchers like Joe Subotnik or Noel Hush, who proposed such unifications in different ways and with different methods.

2) I was sometimes left unsatisfied by the discussion of the results presented by the authors. More specifically, I would have appreciated a more detailed analysis of the Hab values trends reported in Table 1, from a molecular perspective. Such explanations may be trivial for the authors and researchers in the field, but most likely not to the broad readership of Nature Communications.

3) In p.7, the authors discuss Hab and the fact that such matrix elements appear to be independent of the nuclear geometries. They may want to specify that it is a diabatic quantity reflecting the character of electronic states, and therefore by definition less sensitive to nuclear geometries than quantities expressed in the adiabatic representation.

4) I found that the organization and the flow of the text were sometimes not very clear, hampering a proper understanding of the results presented by the authors and how their main findings connect to their conclusions. The last few paragraphs (p.16-17) of the manuscript also suffer from several typographical errors that could have been easily avoided (see list below). Simplifying the structure of the text and maybe leaving some theoretical discussions for the SI might improve the readability of the manuscript if a broad audience is targeted.

Typographical errors:

"Franck-Condon" and "Creutz" are often misspelled in the text.

p.10: "orbtiats"

p.16: "transiton", "avoied", "Supprisingly"

p.17: "requirment", "avoded", "approxiamtion", "nonadaibatic", "demonstrates", "dyanamics", "kinttics", "transtion".

Caption of Figure 4: a definition of the underlying curve is needed for each plot.

Authors:

1) We agree with the reviewers on use of the word "validation" that is not quite suitable for the context. In the revised manuscript, sentences having words "validate" and "validation" are rephrased. For example:

The sentence "The LZ formula has been exploited to predict whether an ET reaction is adiabatic or nonadiabatic; however, experimental validation of the theory itself has not been achieved." is changed to "The LZ formula has been exploited to predict whether an ET reaction is adiabatic or nonadiabatic; however, experimental verification of the

theory becomes a challenge. (see page 2-3)

Page 5: "This work has validated, for the first time, the LZ model and revealed the energetic and dynamic details of a system crossing over the two limits, which are not well described by this model." is changed to "Now, we present the first experimental verification on the LZ model, revealing the energetic and dynamic details of a system crossing over the two limits, which are not well described by this model. With the results from this MV $[\mathbf{EE}'-(\mathbf{ph})_n-\mathbf{EE}']^+$ system, unification of the contemporary ET theories under the semiclassical framework is visualized"

In the abstract, sentence "The results and conclusions unify the contemporary ET theories under the semiclassical framework." is changed to "In this study, the contemporary ET theories under the semiclassical framework are unified with identical outcomes in the intermediate regime."

2) A new figure (Fig. 2 in the new version) is added, which presents the IVCT spectra for two series of compounds. New text is added to briefly introduce donor-acceptor ET in the Mo₂ dimers and optical derivation of the Hab parameter. In the following section, variation trends of Hab for the series are discussed in more detail (see page 8).

3) In this context, a sentence "These results reflect the diabatic nature of the electronic states in these ph₃ systems, in contrast to the adiabatic systems which exhibit the H_{ab} parameters sensitive to nuclear geometry as in the $[\mathbf{EE}'-\mathbf{ph}-\mathbf{EE}']$ series." is added (see page 8).

4) We hope that the discussions are helpful for experimental researchers to get the picture of nonadiabatic transition, because it is not straightforward to understand the physical details due to the lack of experimental studies. Therefore, we wish to keep the discussion of transition channels in the text, if it is okay. The other reason is, it seems to the authors, that with this part included we are telling the complete story.

Thanks a lot for help with checking the spelling. All the typos are corrected.

The figure (now Fig. 5) caption is modified with a definition for each of plot.

Reviewer #2:

The paper investigates the electron transfer (ET) process in a systematic series of bridged metal complexes with the goal of validating the Landau-Zener formula for the electron-transfer rate in the non-adiabatic regime.

The paper starts with an account of the basic ingredients in ET theories. Then strategy followed for the extraction of the various parameters.

The main discussion of the paper at the end of page 13, until page 16, compares then the ET rate constants calculated using Marcus theory, which is valid in the adiabatic regime, and LZ rates, and shows how in some cases they provide divergent results, whereas in the intermediate regime they agree quite well.

My impression is that the strength of the paper is the very systematic comparison of many compounds across the adiabatic to non-adiabatic regimes and the use of two different models, Marcus and LZ theories.

I have a more fundamental problem with this work, namely with its premises and how it frames the achieved results:

In line 48 the authors write: "The LZ formula has been exploited to predict whether an ET reaction is adiabatic or nonadiabatic; however, experimental validation of the theory itself has not been achieved."

I have been wondering, what does this statement exactly mean. If the LZ formula had already been used to predict the adiabaticity or lack thereof, it means that the corresponding parameters (for Eqs. 1a,b,c) could already be extracted from experiments before this work, so this part is not really new.

Then I would ask, had the formula made right or wrong predictions?

Specifically, the paper claims that it "validates" the theory. To me, a theory is validated when its prediction matches experiment. Table I lists Marcus and LZ rate constants, but it does not list experimental ET rates measured independently and directly, i.e. not derived from Eqs. 9 and 11.

So, I wonder what has been validated.

Further down in line 301 one reads:

"Therefore, this work shows that the adiabatic and nonadiabatic ET rate expressions are applicable in the respective ET dynamic limits and work equally well with accordant results for the LZ intermediates, although a single theory that rigorously treats the two limits is not available."

Hence, "validation" of the LZ theory means in this work that in the intermediate regime it coincides with another theory that approaches the rate from the other side, and that each one is good in their respective limits. I think this is a very interesting result, but not a validation of the theory per se.

I would suggest that the authors clarify this point and ideally include measured rate constants in Table I to provide an idea about the accuracy of the two

theories.

Authors:

We agree on the comments of Reviewer #2 about use of the word “validate” or “validation”. Originally, by “validation” we meant that ET energetic, dynamic and kinetic data derived from different experimental methods and theoretical models are in excellent agreement with the variation trends of the LZ parameters that describe the nonadiabatic transition, providing experimental verification or manifestation to the LZ model. There is no intent to say that the LZ model is invalid or needs to be invalidated. But indeed, the semantic confusion exists. In the new version, changes are made accordingly to correct the confusing expressions (see the color highlight lines in the new manuscript). If further clarifications needed, please advise.

Thanks to Reviewer #2 for pointing out the significance of this work, specifically characterization of the intermediate. This view is included in the abstract of the new version, by saying “In this study, the contemporary ET theories under the semiclassical framework are unified with identical outcomes in the intermediate regime.”

In this study, all the physical parameters, including ET rate constants, are optically determined based on the Hush-Marcus theory. There is no measured or observed ket available due to the lack of applicable techniques

Reviewer #3:

This is a well-written manuscript that describes the application of the Landau-Zener formula to describe non-adiabatic transitions within the context of thermal electron transfer. The work is based on a series of mixed-valence complexes consisting of pi-bridge coupled Mo2 dimer units. The experiments are creative and informative, and the topic is highly interesting. The work is likely to be well received and provides a nice explanation of fundamental ET for a wide audience.

The main downside of this paper is the disconnection from the raw spectroscopic data, which would have been better to include at least a couple of representative spectra in the main text, rather than fully relegated to the SI. Perhaps there is a limit on the number of figures available for Nature Communications, but I would strongly encourage the authors to include a figure showing some of the spectra if possible, as it would substantially improve the existing story and also provide a reality check on the level of quantitative detail that may be obtained using this approach.

Authors:

As suggested, a figure (Fig. 2 in the revised version) is added, which presents the IVCT spectra for two series $[\text{EE}'\text{-ph-EE}]^+$ (E, E' = O, S) and $[\text{SS}-(\text{ph})_n\text{-SS}]^+$ (n = 1-3). With these two series as representatives, the variation of spectral characteristics as O/S alternation and ph number change is shown. In the text, spectral features such

as transition energy, band width and shape are described and discussed in terms of electronic coupling degree (see the highlight text in page 7).

Reviewer #3:

The other issue I have with the manuscript is one that is actually a strength, overall. The distinction between diabatic and adiabatic modes of ET is somewhat arbitrary. These are really just two different views of the same thing. For example, in eqns 9 and 11, κ is effectively a measure of H_{ab}^2 . In other words, it seems like a circular argument to compare the diabatic and non-adiabatic rate constants that are derived from the same spectroscopic information. Nevertheless, the discussion surrounding diabatic vs adiabatic ET is interesting and forces a closer understanding of the fundamental aspects of ET. The distinction between the two approaches is interesting as it relates to the mechanism for ET, and whether the electron transfer event is sudden and well defined, or occurs gradually (i.e. adiabatically) along the nuclear coordinate. Moreover, the manuscript gives a clear description of the two limiting cases and the data nicely illustrate trends spanning the range between them.

An easy point of confusion is the difference between the terms adiabatic, diabatic, and non-adiabatic. It would be helpful to a broad audience for the authors to explicitly define each of these in the context that they are used here. For example, clarify the statement in line 45 what it means to have “thermal ET though non-adiabatic transition”. At face value, this seems to be ET that occurs from the ground state to an excited state, but here it actually means an adiabatic ET that remains on the lower potential energy curve throughout the reaction, but with a very weakly avoided crossing (i.e. small H_{ab}). (More specifically, it is unclear what to make of the statement: “when the adiabatic aided crossing is similar to the diabatic crossing” on line 46.) While this may be semantics, a clear description will be important to make this interesting work more accessible to a broad audience.

Authors:

In the revised manuscript, the terms adiabatic, diabatic and nonadiabatic are defined and briefly explained in the context: “By nonadiabatic transition, the system crosses the intersection between the potential energy surfaces (PES) of two diabatic states along the reaction coordinate. Nonadiabatic coupling of the reactant and product states increases the probability of the system traversing the transition state, eventually generating the adiabatic states and leading adiabatic crossover of the system.” (see the highlight text in page 2).

Thanks very much for carefully checking the spelling. All typos are corrected.

REVIEWER COMMENTS

Reviewer #1 (Remarks to the Author):

I thank the reviewer for their answers and additions following my questions 2 and 3.

However, I am left somehow unsatisfied by the answer given by the authors to my first comment. My comment #1 has been addressed by simply removing 'validation' and replacing it with 'verification', while I was challenging the way the authors discussed and presented the Landau-Zener model. I think that the nature of this model should be contrasted with other more advanced theoretical descriptions of electron transfer. In particular, its (known) limitations should be highlighted in contrast to other models. I also struggle to understand the meaning of the new sentence proposed for the abstract: "In this study, the contemporary ET theories under the semiclassical framework are unified with identical outcomes in the intermediate regime." I am not quite sure that I understand what the authors mean by "theories [...] are unified with identical outcomes".

I also found that the following new sentences added to explain the diabatic/adiabatic/nonadiabatic terminology are somewhat confusing:

"By nonadiabatic transition, the system crosses the intersection between the potential energy surfaces (PES) of two diabatic states along the reaction coordinate. Nonadiabatic coupling of the reactant and product states increases the probability of the system traversing the transition state, eventually generating the adiabatic states and leading adiabatic crossover of the system."

Adiabatic and diabatic are two different representations of the electronic states. H_{ab} is a matrix element of the electronic Hamiltonian in the diabatic representation and often called 'diabatic coupling'. Adiabatic states are the eigenvalues of the electronic Hamiltonian and are connected by nonadiabatic coupling terms that emanate from the kinetic energy operator (whose matrix representation is no more diagonal in the adiabatic representation). 'Nonadiabatic transition' is a term often used in the adiabatic representation to express the transfer of nuclear amplitude from one electronic (adiabatic) state to another under the action of nuclear motion. I also wonder if the use of 'transition state' for PES in the diabatic representation is adequate, as diabatic surfaces will always cross and not form per se a transition state. Such a transition state will be observed only in the adiabatic representation, for the lower *adiabatic* PES, as a result of a sizable diabatic coupling. Hence, the sentences proposed by the authors mix the terminology of different electronic representations, which can lead to confusion. The authors may want to reword this part. More details on these terms can be found for example in the book "Conical Intersections, Vol. 15", by Domcke, Köppel, and Yarkony.

Reviewer #2 (Remarks to the Author):

The three referees have raised, to some extent, overlapping concerns that have been addressed in the new version. In particular, the semantic issue with the word "validation" referred to the LZ theory has been solved. The scope and intention of the work are much more clear now.

In my opinion, the concerns of the referees have been correctly addressed, and my recommendation is to publish the work in its current form.

Oriol Vendrell

Reviewer #3 (Remarks to the Author):

The authors made changes to address reviewer comments that are mostly helpful. I particularly appreciate that among these changes the authors have added the new Fig 2 showing two series of representative IVCT "bands". Unfortunately, I have to take issue with the way it is presented in the revised ms as being unclear, or even misleading. Unlike some of the figures in the SI, the new Fig 2 shows only the "simulated" IVCT bands, rather than the actual absorption spectra. This is problematic because it gives a misleading impression of the signal-to-noise. Spectra in the SI reveal that the actual IVCT bands are more difficult to distinguish due to the overlapping metal and MLCT bands, and in some cases are barely seen above the experimental noise level. Although the actual spectra are shown in the SI, it seems critically important to me that a more accurate representation of the actual data is provided in the main text so that the reader understands not only from where the data are extracted, but also some of the experimental limitations. To address this, the figure needs to include at least one (maybe a few) actual spectra in addition to the series of simulated IVCT bands in order to show how the latter are obtained. Alternatively, the the authors could include estimated uncertainties for each of the spectroscopic values (and rate constants) in Table 1 to more rigorously address any experimental variability of observed trends.

Responses to Referees

Reviewer #1:

I thank the reviewer for their answers and additions following my questions 2 and 3. However, I am left somehow unsatisfied by the answer given by the authors to my first comment. My comment #1 has been addressed by simply removing 'validation' and replacing it with 'verification', while I was challenging the way the authors discussed and presented the Landau-Zener model. I think that the nature of this model should be contrasted with other more advanced theoretical descriptions of electron transfer. In particular, its (known) limitations should be highlighted in contrast to other models. I also struggle to understand the meaning of the new sentence proposed for the abstract: "In this study, the contemporary ET theories under the semiclassical framework are unified with identical outcomes in the intermediate regime." I am not quite sure that I understand what the authors mean by "theories [...] are unified with identical outcomes".

Authors:

We thank reviewer #1 for kindly contributing his/her expertise, carefully examining the manuscript and raising important issues. The manuscript has been revised according to the reviewers' comments and suggestions. All the changes are color highlighted. Here we give our responses or explanations to each of the concerns. The words "validation" and "verification" in the manuscript are removed, replaced with "demonstration" and "manifestation"

There are several theoretical models exploited to deal with crossover issue and solve the narrow range problem of the LZ mode, for example, multilevel blocking Monte Carlo simulations under spin-boson model description (J. Chem. Phys. 2003, 118, 179) and the Zhu-Nakamura theory (J. Chem. Phys. 1995, 102, 7448). With our current data, we have made comparison with the data derived from the spin-boson model to probe the crossover regime. In the new approach, at high temperatures, the crossover regime is expected for $\Delta/\omega_c \approx (\Lambda/\omega_c)^{1/2}$, where $\Delta = 2H_{ab}$ and Λ is total reorganization energy. We found that in our Mo2-Mo2 system, this relation is satisfied only for [OO-ph-OO], [OS-ph₂-OS] and [SS-ph₂-SS], which have a Λ/Δ ratio < 10 but > 4 . However, for these systems, the LZ $\kappa_{el} =$ or ≈ 1 , for the LZ intermediates systems [OS-ph₃-OS] and [SS-ph₃-SS], $\Delta/\omega_c \gg (\Lambda/\omega_c)^{1/2}$, as shown in the following Table. Therefore, there exist significant disagreement between the two theoretical approaches. For the current Mo2 mixed valence system, since the LZ formula defined

crossover regime adiabatic and nonadiabatic treatments give nearly identical ET rates, we believe that the LZ formula is more suitable. These results are not presented in the manuscript because of the significant disagreements between the two theory.

Table:

	Δ/ω_c	$(\Lambda/\omega_c)^{1/2}$	$\Delta (2H_{ab})$	$\Lambda (E_{IT})$	Λ/Δ	$\kappa_{el} (LZ)$
[OO-ph-OO]	7.05	5.04	1178	4240	3.6	1
[OS-ph-OS]	8.69	4.54	152	3440	2.4	1
[SS-ph-SS]	10.35	4.01	1728	2640	1.5	1
[OO-ph ₂ -OO]	2.28	7.05	380	8300	22	0.75
[OS-ph ₂ -OS]	4.24	6.26	708	6536	9.2	0.98
[SS-ph ₂ -SS]	4.97	5.38	830	4826	5.8	1
[OO-ph ₃ -OO]	0.74	8.62	124	12400	100	0.14
[OS-ph ₃ -OS]	1.44	6.66	240	7400	31	0.48
[SS-ph ₃ -SS]	1.60	6.09	268	6200	23	0.58

The new sentence in the abstract is:

This study shows that in the intermediate regime, the ET kinetic results derived from the adiabatic and nonadiabatic formalisms are almost identical, in accordance with the Landau-Zener model.

With our experimental results, which show that the LZ formula can be applied in broader range of energy and coupling strength than the theoretical limit, few related sentences are reworded. Please see the highlighted lines.

Reviewer #1:

I also found that the following new sentences added to explain the diabatic/adiabatic/nonadiabatic terminology are somewhat confusing:

"By nonadiabatic transition, the system crosses the intersection between the potential energy surfaces (PES) of two diabatic states along the reaction coordinate. Nonadiabatic coupling of the reactant and product states increases the probability of the system traversing the transition state, eventually generating the adiabatic states and leading adiabatic crossover of the system."

Adiabatic and diabatic are two different representations of the electronic states. Hab is a matrix element

of the electronic Hamiltonian in the diabatic representation and often called 'diabatic coupling'. Adiabatic states are the eigenvalues of the electronic Hamiltonian and are connected by nonadiabatic coupling terms that emanate from the kinetic energy operator (whose matrix representation is no more diagonal in the adiabatic representation). 'Nonadiabatic transition' is a term often used in the adiabatic representation to express the transfer of nuclear amplitude from one electronic (adiabatic) state to another under the action of nuclear motion. I also wonder if the use of 'transition state' for PES in the diabatic representation is adequate, as diabatic surfaces will always cross and not form per se a transition state. Such a transition state will be observed only in the adiabatic representation, for the lower *adiabatic* PES, as a result of a sizable diabatic coupling.

Hence, the sentences proposed by the authors mix the terminology of different electronic representations, which can lead to confusion. The authors may want to reword this part. More details on these terms can be found for example in the book "Conical Intersections, Vol. 15", by Domcke, Köppel, and Yarkony.

Authors:

The original sentence is indeed confusing and does not clearly describe how nonadiabatic transition is achieved. In this revised version, the new sentence is:

By nonadiabatic transition, electron transfer proceeds adiabatically crossing the intersection between the reactant and product potential energy surfaces (PES), while the simultaneous and instantaneous transfer of nuclear amplitude between the two adiabatic states takes place nonadiabatically under the action of nuclear motion. Nonadiabatic coupling of the reactant and product diabatic states increases the probability of system traversing the crossing point, eventually leading to electron transfer in the adiabatic limit.

Reviewer #2:

The three referees have raised, to some extent, overlapping concerns that have been addressed in the new version. In particular, the semantic issue with the word "validation" referred to the LZ theory has been solved. The scope and intention of the work are much more clear now.

In my opinion, the concerns of the referees have been correctly addressed, and my recommendation is to publish the work in its current form.

Reviewer #3:

The authors made changes to address reviewer comments that are mostly helpful. I particularly appreciate that among these changes the authors have added the new Fig 2 showing two series of representative IVCT "bands". Unfortunately, I have to take issue with the way it is presented in the revised ms as being unclear, or even misleading. Unlike some of the figures in the SI, the new Fig 2 shows only the "simulated" IVCT bands, rather than the actual absorption spectra. This is problematic because it gives a misleading impression of the signal-to-noise. Spectra in the SI reveal that the actual IVCT bands are more difficult to distinguish due to the overlapping metal and MLCT bands, and in some cases are barely seen above the experimental noise level. Although the actual spectra are shown in the SI, it seems critically important to me that a more accurate representation of the actual data is provided in the main text so that the reader understands not only from where the data are extracted, but also some of the experimental limitations. To address this, the figure needs to include at least one (maybe a few)

actual spectra in addition to the series of simulated IVCT bands in order to show how the latter are obtained. Alternatively, the authors could include estimated uncertainties for each of the spectroscopic values (and rate constants) in Table 1 to more rigorously address any experimental variability of observed trends.

Authors:

Thanks for the suggestions. Figure 2 now is replaced with original absorption bands for the two series complexes. The only treatment to the original spectra is to trim the overtones in IR region, with the band profile remaining unchanged. For each of the complexes parallel measurements have been carried out and experimental errors are expressed with mean deviations, as presented the Table 1. Although measurements and Gaussian simulations of the band profiles generate appreciable deviations, there are no significant experimental errors introduced to the rate constants.

REVIEWERS' COMMENTS

Reviewer #1 (Remarks to the Author):

I thank the authors for their answers.

I would suggest:

"Coupling between the diabatic states of the reactant and the product..." instead of

"Nonadiabatic coupling of the reactant and product diabatic states..." (nonadiabatic coupling is a terminology related to the adiabatic representation.

The last sentence of the manuscript should be checked for typographical errors ("demonstrate" and "usuful").

Reviewer #3 (Remarks to the Author):

The new text on page 2 (line 36-41) in response to reviewer #1 does not make any sense. This needs to be fixed in order to accurately (and clearly) explain the difference between adiabatic and non-adiabatic processes. This point is closely related to a concern I raised in my first review, and now has been made even less clear in the latest version of the manuscript. Reviewer #1 provided a very clear explanation in their second review that should provide a useful framework for the authors to clarify this point. An article that claims to "verify" (or even to "demonstrate") the L-Z model really needs to explain adiabatic and non-adiabatic processes in a clear and understandable manner. Anything less would not be suitable for a journal with a broad audience or of the caliber of Nature Communications.

The new Fig 2 is much more informative. This version provides the reader a chance to see how information is extracted from the experimental IVCT bands. The "cut-off" region is not well explained, but (maybe?) it is not a critical issue. The figure does raise some questions (e.g., why are there not any vibrational bands observed in the spectra?) but at least the reader can evaluate the strength (or weakness) of the data used in the analysis.

The uncertainties in Table 1 are a valuable addition, although the authors really should explain where they come from. The table footnote only indicates that they “are the mean deviations”, but mean deviations of how many measurements? And are they independent measurements? In their “responses to referees”, the authors suggest that these come from “parallel measurements” ... does this mean 2 measurements of each compound? These are important details that need to be shared with readers in order to allow critical evaluation. The mean deviation between 2 values is VERY different from an estimated statistical uncertainty!

Responses to the referees

Many thanks to the reviewers for examining the manuscript. With their help, we are able to complete the three revised versions of the manuscript, significantly improving the quality of the article. Here we give "point by point" responses to the reviewers' comments. In the revised manuscript, all the changes made on the last version are yellow highlighted.

Reviewer #1:

I would suggest:

"Coupling between the diabatic states of the reactant and the product..." instead of

"Nonadiabatic coupling of the reactant and product diabatic states..." (nonadiabatic coupling is a terminology related to the adiabatic representation.

The last sentence of the manuscript should be checked for typographical errors ("demonstrate" and "usuful").

Authors:

Change to the sentence has been made. The typos are corrected.

We all add definitions on adiabatic and nonadiabatic limits of ET, as suggested by Reviewer #3. Please check with your expertise.

Reviewer #3:

The new text on page 2 (line 36-41) in response to reviewer #1 does not make any sense. This needs to be fixed in order to accurately (and clearly) explain the difference between adiabatic and non-adiabatic processes. This point is closely related to a concern I raised in my first review, and now has been made even less clear in the latest version of the manuscript. Reviewer #1 provided a very clear explanation in their second review that should provide a useful framework for the authors to clarify this point. An article that claims to "verify" (or even to "demonstrate") the L-Z model really needs to explain adiabatic and non-adiabatic processes in a clear and understandable manner. Anything less would not be suitable for a journal with a broad audience or of the caliber of Nature Communications.

Authors:

We agree. Definitions on the adiabatic and nonadiabatic limits of ET are given in this revised version (see the lines highlighted on page 2). In the last version, we modified the definition of nonadiabatic transition (line 36-41) according to Reviewer #1's suggestion, which is kept in this revision.

Reviewer #3:

The new Fig 2 is much more informative. This version provides the reader a chance to see how information is extracted from the experimental IVCT bands. The "cut-off" region is not well explained, but (maybe?) it is not a critical issue. The figure does raise some questions (e.g., why are there not any vibrational bands observed in the spectra?) but at least the reader can evaluate the strength (or weakness) of the data used in the analysis.

The uncertainties in Table 1 are a valuable addition, although the authors really should explain where

they come from. The table footnote only indicates that they “are the mean deviations”, but mean deviations of how many measurements? And are they independent measurements? In their “responses to referees”, the authors suggest that these come from “parallel measurements” does this mean 2 measurements of each compound? These are important details that need to be shared with readers in order to allow critical evaluation. The mean deviation between 2 values is VERY different from an estimated statistical uncertainty!

Authors:

The “cut-off” phenomenon in mixed-valence chemistry is very important, which reflects the donor-acceptor coupling strength. In this work, variation of cut-off area in a series of dyads corresponds to the change of Hab parameters, as expected. Theoretically, it is a clear representation of vibronic transition under Franck-Condon principle and well understood on the basis of Boltzmann population of vibrational levels within the two-state mode. These understandings have been established through complex and organic mixed-valence systems. In this article, to focus on the nonadiabatic transition issue, we did not invoke discussion on the shape of spectra.

For each of the nine Mo₂ dimers, at least three independent parallel measurements were carried out; some of the complexes have been measured five times, giving the mean deviations in the parentheses in Table 1 in the last version of the manuscript. This is because these compounds have been investigated at different period of time spanning years. Now we do the deviation analysis using the original data taken at different time. For some compounds which lacked data for analysis we also conducted new measurements. Actually, not only were the data shown in this paper not collected at the same time, but also not on the same instrument. For strong coupling systems with the IVCT bands extending to the IR region we need to combine the data obtained from two different spectrometers that cover the Visible-near IR (300-3000 nm) and IR (3333-2000 cm⁻¹) regions, respectively. For the weakly coupled compounds which have high energy absorptions only a UV-3000 nm spectrometer is needed. In the spectra few overtones are observed in the higher energy region (< 3000 nm), but the spectra measured on the IR instrument are overlapped with dense vibrational bands, but the IVCT profiles are clearly seen. And also, for measurements of each wavelength range, different instruments had been used due to the instrumental availability at that time. Spectra recorded on different instruments may show some differences in overtone or noise display. In the SI, the original spectra show the vibrational bands and overtones, which are trimmed to outline the spectral shape in Fig. 2. In this revision, detailed spectroscopic methods are added in the supplementary information. We think that using data measured at different time and on different methines for error analysis is even more trustable.

In this revision, we decide to use standard deviations for the ET kinetic data (ket), while averaged data are presented for the extracted optical data and the derived physical parameters (Table 1). In this work, the calculated standard deviations are generally close to or smaller than the mean deviations. Considering that the experimental errors, including spectrum recording and data extraction, do not cause significant deviations to the kinetic data and finally do not affect the LZ analysis results, we prefer to show the uncertainty only to rate constant and electronic transition frequency (ν_{el}). The resultant uncertainties to the LZ parameters (γ , P_0 , and κ_{el}) are negligible.